# National mapping of schistosomiasis, soil-transmitted helminthiasis and anaemia in Yemen: Towards better national control and elimination

**Nur Alia Johari**[1¤a‡], **Dhekra Amin Annuzaili**[1¤b‡]*, **Hani Farouk El-Talabawy**[2], **Maryam Ba-Break**[3], **Abdulsalam M. Al-Mekhlafi**[4], **Samira Al-Eryani**[4], **Abdulhakim Ali Alkohlani**[5¤c], **Albis Francesco Gabrielli**[6], **Riadh Ben-Ismail**[6], **Sami Alhaidari**[7], **Adel Muaydh**[7], **Rasheed Alshami**[7], **Magid Al Gunaid**[7], **Alaa Hamed**[5], **Nehad Kamel**[5], **Karen Palacio**[8], **Fiona Fleming**[1], **Michael Duncan French**[1¤d]

1 Schistosomiasis Control Initiative Foundation, London, United Kingdom, 2 Regional Office for the Eastern Mediterranean, Department of Information, Evidence and Research, Department of Communicable Diseases, World Health Organization, Cairo, Egypt, 3 Nuffield Centre for International Health and Development, University of Leeds, Leeds, United Kingdom, 4 Department of Parasitology, Faculty of Medicine and Health Sciences, Sana'a University, Sana'a, Yemen, 5 The World Bank, Washington D.C, United States of America, 6 Regional Office for the Eastern Mediterranean, Department of Communicable Diseases, World Health Organization, Cairo, Egypt, 7 The National Schistosomiasis Control Programme (NSCP), Department of Primary Health Care, Ministry of Public Health and Population, Sana'a, Yemen, 8 The End Fund, New York, New York, United States of America

¤a Current address: Institute for Research, Development, and Innovation (IRDI), International Medical University, Kuala Lumpur, Malaysia
¤b Current address: The National Schistosomiasis Control Programme (NSCP), Department of Primary Health Care, Ministry of Public Health and Population, Sana'a, Yemen
¤c Current address: Yemen Field Epidemiology Training Program, Sana'a, Yemen
¤d Current address: Research Triangle Institute (RTI) International, Washington D.C, United States of America
‡ These authors share first authorship on this work.
* dannuzaili@gmail.com

**Data Availability Statement:** All relevant data are within the manuscript and its Supporting Information files.

## Abstract

### Background

Schistosomiasis (SCH) and soil-transmitted helminthiasis (STH) are known to be endemic in Yemen. However, the distribution of both diseases had not previously been assessed by a well-structured national mapping study covering all governorates. The main aim of this study was, therefore, to map the prevalence of SCH and STH in Yemen in order to better inform implementation of effective national control and elimination interventions. The assessment of the distribution of anaemia was also included as a well-known consequence of infection with both SCH and STH. Secondarily, the study aimed to provide a broad indication of the impact of large-scale treatment on the distribution of infection.

### Methodology and principal findings

To achive these aims, 80,432 children (10–14 years old) from 2,664 schools in 332 of Yemen's 333 districts were included, in 2014, into this national cross-sectional survey.

**Funding:** This study benefited from the financial support of the World Bank, Grant Award No. IDA-H5420 (AH, NK) in partnership with The End Fund (KP) and the Schistosomiasis Control Initiative (NAJ, DAA, FF, MDF, based at Imperial College London at that time). The funders had no role in study design, data collection and analysis, decision to publish, or preparation of the manuscript.

**Competing interests:** The authors have declared that no competing interests exist.

Countrywide, 63.3% (210/332) and 75.6% (251/332) of districts were found to be endemic for SCH and STH respectively. More districts were affected by intestinal than urogenital SCH (54.2% and 31.6% respectively). SCH infection was mostly mild and moderate, with no districts reporting high infection. One quarter (24.4%) of Yemeni districts had high or moderate levels of *Ascaris lumbricoides* infection. Infection with *Trichuris trichiura* was the second most common STH (44.9% of districts infected) after *A. lumbricoides* (68.1%). Hookworm was the least prevalent STH (9.0%).

Anaemia was prevalent in 96.4% of districts; it represented a severe public health problem (prevalence $\geq$ 40%) in 26.5% of districts, and a mild to moderate problem in two thirds of the districts (33.7% and 36.1% respectively).

## Conclusion

This study provided the first comprehensive mapping of SCH, STH, and anaemia across the country. This formed the basis for evaluating and continuing the national control and elimination programme for these neglected tropical diseases in Yemen.

### Author summary

Schistosomiasis (SCH) and soil-transmitted helminthiasis (STH) are endemic in Yemen. However, the precise prevalence and distribution of these diseases had remained unclear as cases were mainly identified sporadically in health facilities when patients with chronic complications sought treatment. This study provided the first comprehensive mapping of SCH and STH infections and anaemia through implementing a countrywide cross-sectional school-based survey. Country-wide, both diseases were found to present a significant public health burden. SCH and STH prevalence were more than 60.0%, and more districts were affected by intestinal than urogenital SCH. Based on the WHO classification, the SCH levels of infection ranged from mild to moderate. One quarter of Yemeni districts had severe or moderate levels of *A. lumbricoides* infection. Trichuriasis is the second most common STH after Ascariasis, with a mild to moderate level of infection in most districts, except three districts where it was severe. Anaemia is a severe or moderate public health problem in 26.5% and 36.1% of districts respectively. To the best of our knowledge this study presents the first national picture of anaemia burden in school-aged children in Yemen. The study's findings and recommendations will be used to inform the design of the ongoing national control and elimination programmes of these neglected tropical diseases in Yemen.

## 1. Introduction

Schistosomiasis (SCH; also known as Bilharzia), and Soil-Transmitted Helminthiasis (STH) are classified by the World Health Organization (WHO) as neglected tropical diseases (NTDs). Both are widely distributed parasitic diseases that infect hundreds of millions of people worldwide; particularly those who have limited access to safe water, sanitation services, and health facilities. Children of school age typically harbour the highest levels of infection [1–3].

Historical records suggest that SCH is endemic in Yemen with an estimated burden of at least three million cases nationwide at the end of the 20[th] century [4]. In Yemen, SCH is caused

by *Schistosoma mansoni* and *S. haematobium*, digenetic blood trematodes that result in intestinal and urogenital infections, respectively, due to their characteristic oviposition sites. The clinical manifestations of intestinal SCH include blood in the stool, diarrhoea, abdominal pain, ascites, and complications related to chronic damage to the liver and kidneys. Urogenital SCH is associated with blood in urine, bladder and ureter fibrosis, damage to the female reproductive tract (female genital SCH), bladder cancer, and increased risk of HIV transmission [5, 6].

Before large-scale control programmes commenced in Yemen in 2009, over half a million individuals were reportedly suffering from significant clinical morbidity, including hepatosplenomegaly in more serious cases and anaemia, stunting, and poor cognitive development, particularly in young children [4, 7]. The common STH infections—hookworm, *Ascaris lumbricoides* and *Trichuris trichiura*—are also thought to be highly prevalent in Yemen, and are often detected as co-infections with SCH [4, 8]. There had been several historical surveys conducted at the sub-national level in Yemen. For example, Nagi and colleagues demonstrated that both urogenital and intestinal SCH were highly prevalent in Taiz governorate and Khamer district [4, 8]. Another cross-sectional study estimated the prevalence of intestinal parasites among schoolchildren in one remote district, Sahar [9]. However, no national survey had been implemented before the one described in this paper.

Until 2008, treatment against SCH and STH was sporadic, at local scale, and dependent on demand at health facilities and local private pharmacies, in addition to targeting schools in high endemic areas associated with on-the-spot parasitological investigation and treatment. In 2008, a national scale treatment programme against SCH and STH was formulated, led by the Yemen Ministry of Public Health and Population (MoPHP), with funding from the World Bank (WB) and The END Fund, and technical support from the Schistosomiasis Control Initiative (SCI) and the WHO [10]. This programme treated approximately two million school-aged children in its first year (2008) via preventive chemotherapy using the anthelminthic drug praziquantel (PZQ), to which albendazole (ALB) or mebendazole (MEB) were added for STH. Later in 2010, the programme was expanded to include both schools and communities nationwide. Based on historical records, in 2008 the programme started targeting the areas known to be of the highest risk of infection. At the beginning of the programme, epidemiological mapping was undertaken to identify SCH endemic districts across the country with the purpose of stratifying them according to level of risk to start treatment [10]. This was conducted in 2010 and aimed to estimate SCH prevalence using a geospatial model that combined parasitological surveys that sampled students from 108 schools in 27 districts and relevant ecological co-variates. However, the parasitological surveys were selected from only nine (of 23) Yemeni governorates and following a different methodology to the current study, making direct comparisons difficult.

As per the WHO recommendations [11] re-assessment should be implemented after five to six rounds of treatment to determine the progress and thus the next phase of such a large-scale deworming programme. Accordingly, the Yemen MoPHP, supported by the World Bank, the END Fund, WHO, and the SCI, conducted a nationwide prevalence mapping (PM) survey in 2014. This paper presents the findings.

The PM aimed to assess the epidemiology of SCH (*S. mansoni* and/or *S. haematobium*) and STH across all 23 Yemeni governorates following three years of implementation of control interventions by the Yemen national SCH control programme (NSCP). A national 'Yemen Prevalence Mapping Protocol' [12], targeting school-aged children according to WHO guidelines [11], was prepared by the SCI, and applied to the PM survey, with an overall purpose of informing the future national control approach.

The PM survey also included measuring haemoglobin concentration in the same school-aged children, as part of an integrated survey, in what proved to be the first nationwide

anaemia prevalence survey in this population group. In 2013, the Yemen National Health and Demographic Survey (YNHDS) had assessed prevalence of anaemia only among children under five years of age and women in reproductive age (15–49 years old) [13]. Anaemia is a known sequela of various infections prevalent in the country, including SCH, STH, and malaria [14, 15]. The high rates of undernutrition (stunting and wasting) among children, especially in rural populations, [13] makes the countrywide assessment of anaemia crucial.

Accordingly, this nationwide mapping survey was implemented to achieve the objectives of: (1) identifying district-level prevalence of SCH and STH to guide future mass treatment interventions; (2) estimating the level of anaemia prevalence among school-aged children in Yemen in order to make a case for action; (3) evaluating the progress the programme had made against SCH compared to the situation in 2010 and to potentially shift from an objective of control to elimination. However, we note that the 2010 and 2014 surveys used different methodologies which hinders making direct comparisons between them. This article discusses the methods and main findings of this PM survey.

## 2. Methods

### 2.1. Ethics statement

Ethical approval was granted by the Imperial College Ethical Review Board and the ethics committee at the Yemeni MoPHP. Permission to collect data was obtained from governorate and district health and education officials and school managers. Detailed information was provided to parents and the communities in advance, via governorate and district health and education officials, to ensure acceptability of the survey and testing.

Permission to access schools was obtained from the local education authorities. School head-teachers, as gatekeepers, informed all teachers, pupils and their parents/guardians about the study, then individual and independent informed consents were obtained from all participants. Permission was obtained from parents/guardians regarding their child's involvement in the survey. Schools applied an opt-out approach, which is commonly utilised for obtaining consent in school-based research [16–19]: head teachers of all sampled schools contacted parents/guardians to inform them about the survey and asked them to inform the school if they did not permit their child to participate in the survey. Additionally, children were encouraged to discuss their contribution in the study with their parents/guardians before participating in the survey. Moreover, children were asked for their own permission to participate in the study. As recommended in the literature on conducting research in children [17–20], children who refused to participate or whose parents/guardian refused permission were excluded from this study.

All procedures followed during the study were in accordance with the ethical standards of the Helsinki Declaration (1964, 2008) of the World Medical Association. All individuals found to be infected with SCH were provided praziquantel (40 mg/kg body weight).

### 2.2. Study design

A countrywide cross-sectional school-based survey was conducted in 2,664 schools in 332 health districts between February to May 2014. Yemen is divided into 23 governorates (the first administrative unit) which are then further subdivided into 333 districts. Only one district (Wald Rabi' district from Al Bayda' governorate) was excluded from this survey because of its unstable security situation.

The mapping strategy was formulated following WHO guidelines [11, 21]. The recommended sample of 200–250 school-aged children to be drawn from each ecologically homogenous area within the country was applied to each district; this was done with the purpose of

having a finer representation of the burden and distribution of schistosomiasis across Yemen, and also to reflect the fact that the district is considered the 'implementation unit' at which it is logistically feasible to tailor the treatment strategies effectively [22].

## 2.3. Study population and sampling

Eighty thousand four hundred and thirty-two children aged 10 to 14 years were selected from 2,664 schools, following WHO guidelines for evaluating helminth control programmes [11]. The sample size was calculated using EPI Info-7 statistical package [23] using 50.0% as the expected prevalence of SCH and a 0.75% error margin, to ensure a sufficient sample [22]. In each district eight schools were selected using simple random methods, amongst schools that were functioning and included sufficient numbers of both male and female students, as the WHO guideline requests [11, 21]. At least 35 children were selected, using simple random methods, per school to make a total of 280 children sampled per district [12] as per the WHO guideline [11, 21].

## 2.4. Data collection

Data were collected by parasitological technicians (n = 125) forming 25 teams (5 parasitological technicians per team) that were recruited and trained on sample collection techniques and precautionary measures at Sana'a University under the guidance of professors from the Department of Parasitology. The teams were given a manual for field data collection to remind them of the training content. Targeted districts were allocated to each team based on logistics and feasibility.

Upon arrival at each school, teams randomly selected students after informed written consent was obtained from school headteachers, and oral assent was received from the children. Each student was provided with empty stool and urine containers and instructed on how to collect a sufficient amount of each sample. Stool samples collected from each student were tested for the presence of *S. mansoni* and the STHs (*Ascaris*, *Trichuris* and hookworm) using the Kato-Katz method [24, 25]. Urine samples were tested for *S. haematobium* with a two-step method that employed urine filtration and haematuria reagent strips (Hemastix dipsticks) in order to increase the specificity of testing and minimize false-positive results due to potential samples contamination with blood from menstruation or other infections [24, 25]. In order to determine the prevalence and severity of anaemia, the haemoglobin concentration was assessed using HemoCue and the cut off point for identifying the anaemia was adapted to the altitude of each school, following the international guidelines [26–28]. The coordinates and altitude of all sampled schools were recorded with a Global Positioning System (GPS) (eTrex, Garmin International, Kansas, U.S.A) at arrival and at departure. All data and information collected were noted on prepared case report forms.

For quality assurance, supervisory field visits were conducted throughout the data collection period, to ensure the teams followed the correct methodology and processes during sampling. Additional random sub-samples (10%) of Kato-Katz slides from each school were re-examined by a second member of the team blinded to the initial results. Results for the two readings of the slides were compared for any discrepancy [10]. Moreover, weekly progress reports and results were collected and monitored by supervisors at Sana'a University, the manager of the control programme and the SCI.

## 2.5. Data analysis

Data on paper case report forms were entered into Microsoft Excel and sent via email to the university team in Sana'a along with the original hard copies. The results were then cross-

checked before being sent to the SCI. Cleaning and consistency checks were conducted, and average prevalence of infection with *S. mansoni*, *S. haematobium*, *A. lumbricoides*, *T. trichiura*, hookworm as well as prevalence of mild, moderate, and severe anaemia were calculated, along with crude confidence intervals at governorate and district level. Robust 95% confidence intervals (CIs) were generated using the statistical software R 3.1.1 (R Foundation for Statistical Computing, Vienna, Austria) with the R 'survey' package (T. Lumley (2012) "survey: analysis of complex survey samples". R package version 3.28–2). District-level CIs were calculated accounting for clustering at the school-level, where each child was not considered an independent data point with the assumption that children from the same school would show less heterogeneity than those from different schools.

Maps of the prevalence of infection by district and governorate level were developed using QGIS 2.4.0-Chugiak (Open-Source Geospatial Foundation Project). The geographical information system (GIS) data for Yemen was obtained from the DIVA-GIS database (www.diva-gis.org/gdata) and the Yemen MoPHP (http://www.mophp-ye.org/english/data.html). Yemeni districts were classified into high, moderate, or low risk categories for SCH and STH based on the cut-offs detailed in Tables 1 and 2 respectively [11, 12]. Each district was also further classified by the upper boundary of the 95% confidence intervals, where districts with borderline prevalence rates or upper limits that overlap could be identified and warrant more attention from the control programme. Additional maps of the estimated number of infected cases in the country were developed using the MoPHP population data and calculated prevalence rates. In this map, the population size [29], was projected to expected levels in 2014 based on medium-variant projections by the United Nations and MoPHP [30].

Following WHO recommendations, data on haemoglobin concentrations were used to classify cases of anaemia into mild, moderate, or severe categories as shown in Table 3. Haemoglobin concentrations were adjusted for altitude, with the assumption that communities living at high altitude (>1000m above sea level) have generally higher haemoglobin concentrations than those living at lower altitudes [31].

To identify the public health significance of anaemia, the district-level prevalence of anaemia was used to classify districts into severe ($\geq$40%), moderate (20.0–39.9%), mild (5.0–19.9%), or non-anaemic ($\leq$4.9%) categories, as outlined by the WHO [32].

**Table 1. Classification of the risk for schistosomiasis based on the WHO cut-off values of district-level prevalence.**

| Category * | Thresholds for any form of schistosomiasis |
|---|---|
| High-risk | $\geq$50% |
| Moderate-risk | $\geq$10% and <50% |
| Low-risk | <10% |

* Categories and cut-off values listed based on WHO guidelines [11].

**Table 2. Classification of the risk for STHs based on the WHO cut-off values of district-level prevalence.**

| Category * | Cumulative prevalence of all STH | The % of moderate/heavy intensity infections |
|---|---|---|
| High-risk | $\geq$50% | $\geq$10% |
| Moderate-risk | $\geq$20% and <50% | <10% |
| Low-risk | <20% | <10% |

* Categories and cut-off values listed based on WHO guidelines [11].

**Table 3. Classification of anaemia severity at sea level based on measured haemoglobin concentrations by age group.**

| Age group | | Non-Anaemia* (g/L) | Anaemia* (g/L) | | |
|---|---|---|---|---|---|
| | | | Mild | Moderate | Severe |
| 6–59 months | | ≥110 | 100–109 | 70–99 | <70 |
| 5–11 years | | ≥115 | 110–114 | 80–109 | <80 |
| 12–14 years | | ≥120 | 110–119 | | |
| ≥15 years | Girls** | ≥120 | 110–119 | | |
| | Boys | ≥130 | 110–129 | | |

*WHO recommended anaemia cut-offs for measured haemoglobin levels [31–33].

** Non-pregnant girls.

## 3. Results

This study included a total of 80,432 children aged 10–14 years drawn from 2,664 schools in 332 districts from all the Yemeni governorates. Approximately 61% of the samples were male due to poor compliance among female students in various schools. The mean age of participants was 12.5 (inter-quartile range (IQR): 9–11.5 years). All participants were screened for all parasites and anaemia.

### 3.1. Distribution of infection

Of the 332 districts, 63.3% (n = 210) were found to be endemic above the treatment threshold for SCH and 75.6% (n = 251) for STH. This equates to approximately 286,114 infected pupils and 5,510,818 of the general population warranting MDA for SCH or STH. These numbers were calculated based on the UN population projection in 2014 [30]. The guideline for controlling SCH recommends treating school-aged children in districts with low or moderate infection and all populations (school-aged children and adults) in districts with high infection. The Yemen target population was decided on the basis of SCH epidemiology, with ALB/MEB added to the same population.

**3.1.1 Prevalence of schistosomiasis.** Fig 1A and 1B describe the point prevalence of intestinal (*S. mansoni*) and urogenital (*S. haematobium*) SCH infection per district. The findings identified that 31.6% and 54.2% of Yemeni districts were endemic for *S. haematobium* and *S. mansoni* respectively. More districts were affected by intestinal (n = 180) than urogenital (n = 105) SCH. As Table 4 demonstrates, more than half (54.2%) of the Yemeni districts were classified as having either low (48.2%) or moderate (6.0%) risk for intestinal SCH infection. Additionally, about a third (31.6%) of districts were classified as having either low (31.3%) or moderate (0.3%) risk for urogenital SCH infection. In 2014, no district had high risk of either urogenital or intestinal SCH as the prevalence of any SCH was less than 50% in all districts.

Fig 2A and 2B demonstrate the point prevalence of any SCH infection at the governorate and district level. No districts had more than 50% of its population of school-aged children infected by any SCH, so none were classified as having high risk of SCH infection. There was no significant sex difference in the prevalence of SCH.

**3.1.2 Prevalence of Soil-transmitted Helminthiasis.** The findings showed that three-quarters (75.6%) of the Yemeni districts were endemic for any STH infection (n = 251). Fig 3A describes the district-level point prevalence of STH in each Yemeni district. More districts were affected by Ascariasis (n = 226) and Trichuriasis (n = 149) than by hookworm infections (n = 30). Table 4 provides the number of districts affected by each species, and species-specific maps are provided in the supplementary information.

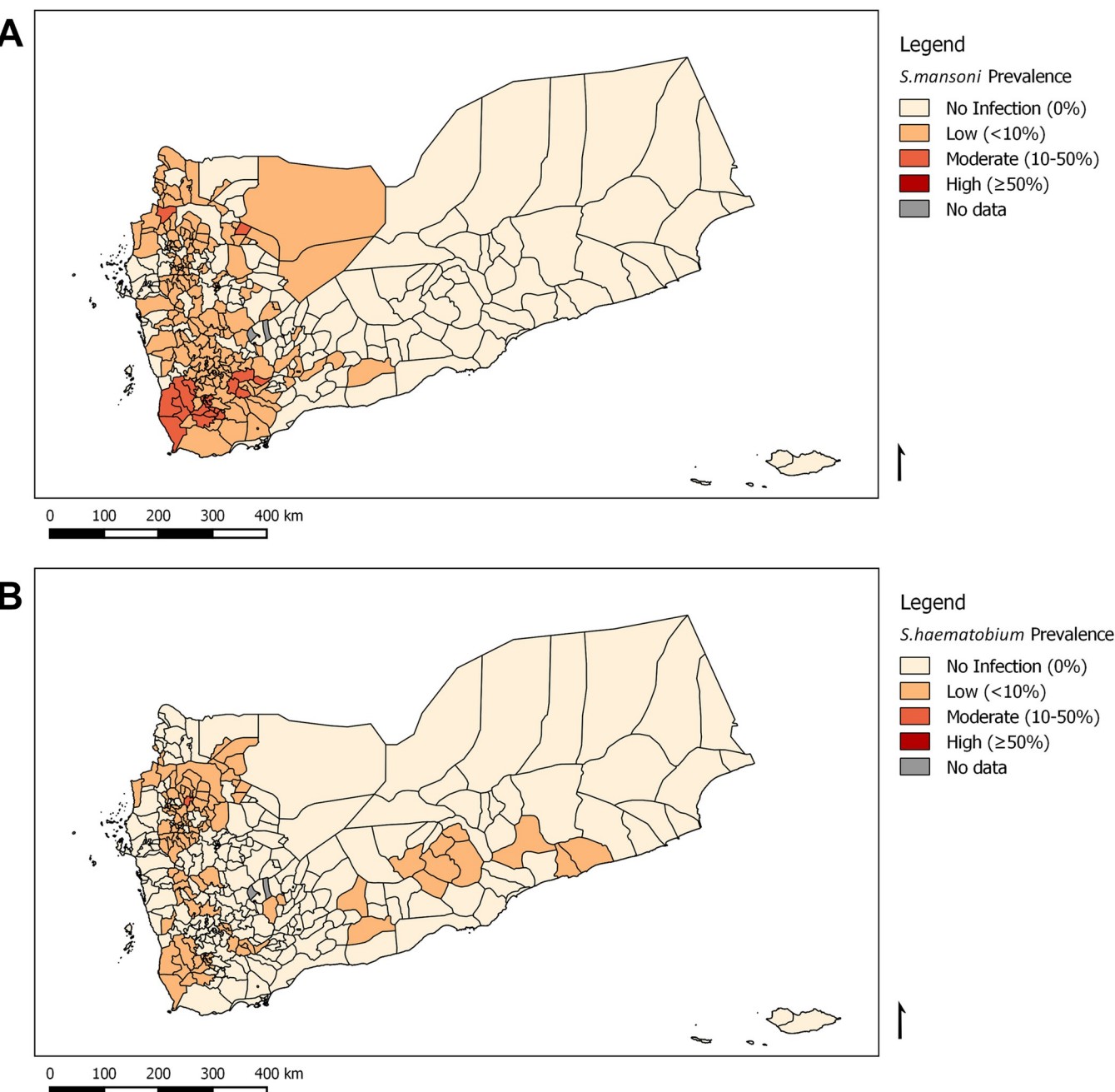

**Fig 1. District-level prevalence maps of schistosomiasis infection in Yemen, 2014.** (A) Intestinal (*Schistosoma mansoni*) and (B) Urogenital (*S. haematobium*) schistosomiasis. These figures were created for this manuscript in QGIS using open source data from DIVA-GIS for the base layers. (DIVA-GIS-http://www.diva-gis.org/gdata).

About two-thirds (65.4%) of the Yemeni districts were classified as having either low or moderate risk of *Ascaris* infection. Fig 3B shows the district-level point prevalence of *Ascaris* infection at the district level. About a fifth (21.7%) of Yemeni districts have *Ascaris* infections among 20%-50% of schoolchildren. Based on the WHO classification [12, 21], nine districts are in the high-risk category, with more than 50% of schoolchildren infected with *Ascaris* (Table 4). Trichuriasis was the second most common STH: 133 districts (40.1%) were classified

**Table 4. The distribution of districts according to risk category for schistosomiasis and soil-transmitted helminth, Yemen, 2014.**

| Risk category | The proportion of districts (Number) | | | | | | | |
|---|---|---|---|---|---|---|---|---|
| | *S. mansoni* | *S. haematobium** | *Ascaris* | | *Trichuris* | | Hookworm | |
| | | | Based on Point Prevalence** | Based on Upper 95% CI*** | Based on Point Prevalence** | Based on Upper 95% CI *** | Based on Point Prevalence** | Based on Upper 95% CI*** |
| High | 0 | 0 | 0.6% (2) | 2.7% (9) | 0 | 0.9% (3) | 0 | 0 |
| Moderate | 6.0% (20) | 0.3% (1) | 11.4% (38) | 21.7% (72) | 1.5% (5) | 3.9% (13) | 0 | 0.3% (1) |
| Low | 48.2% (160) | 31.3% (104) | 56.0% (186) | 43.7% (145) | 43.4% (144) | 40.1% (133) | 9.0% (30) | 8.7% (29) |
| Uninfected | 45.8% (152) | 68.4% (227) | 31.9% (106) | | 55.1% (183) | | 91.0% (302) | |

* Classification based on WHO recommended thresholds (≥50% High-risk, ≥ 10% and <50% Moderate-risk and <10% Low-risk).

** Classification based on WHO recommended thresholds (≥50% High-risk, ≥20% and <50% Moderate-risk and <20% Low-risk) [11]. Districts were classified based on the point prevalence of each STH separately by district.

*** Districts were classified based on the upper boundary of the 95% confidence interval calculated for the prevalence of each STH by district.

as having low levels of *Trichuris*, while sixteen (4.8%) had either high or moderate infection. For both *Ascaris* and *Trichuris*, these districts tended to be poor and rural, with agriculture as the main source of income. Infection was higher in females than males. Hookworm was the least prevalent STH in Yemen, with only 9.0% of districts with infections, all of which were classified under the low-risk category except one moderate-risk district, indicating that the prevalence was less than 20% among schoolchildren (Table 4).

### 3.2. Prevalence and severity of Anaemia

Anaemia is prevalent among schoolchildren in 96.4% of the Yemeni districts. Fig 4 demonstrates the severity of anaemia at the district and governorate levels. Only 12 districts have anaemia in less than 5% of schoolchildren. These districts tended to be urban and had the highest income.

Anaemia is a severe public health problem in over a quarter (26.5%) of Yemeni districts as more than 40% of schoolchildren in these districts had anaemia. Over a third of Yemeni districts (36.1%) had anaemia in 20–40% of schoolchildren, indicating a moderate public health problem based on the WHO classification [31–33]. As shown in Table 5 below, only one third (33.7%) of districts have anaemia in less than one-fifth of schoolchildren and thus have mild prevalence of anaemia. Lack of historical district-level anaemia data limited our ability to estimate the impact of the programme on anaemia. There was no significant sex difference in the prevalence of anaemia (31.0% and 30.1% among females and males, respectively).

### 4. Discussion

This study identified that 63.3% and 75.6% of Yemeni districts are endemic above treatment thresholds for SCH and STH respectively. This represents a reduction in the number of districts estimated to harbour any SCH infection and, importantly, those with high and medium levels of infection compared to 2010 when SCH was known or suspected to be endemic in 82.6% (275/333) of Yemeni districts and STH infection was suspected in all 333 districts [34, 35]. Based on a threshold of ≥40% for SCH, 51 districts were classified as high risk in 2010 compared to the 2014 survey where none of the districts reported SCH prevalence of more than 40%. However, it was noted that if a more conservative high-risk cut-off of ≥30% is utilised, three districts will fall into the high risk category in 2014 [36].

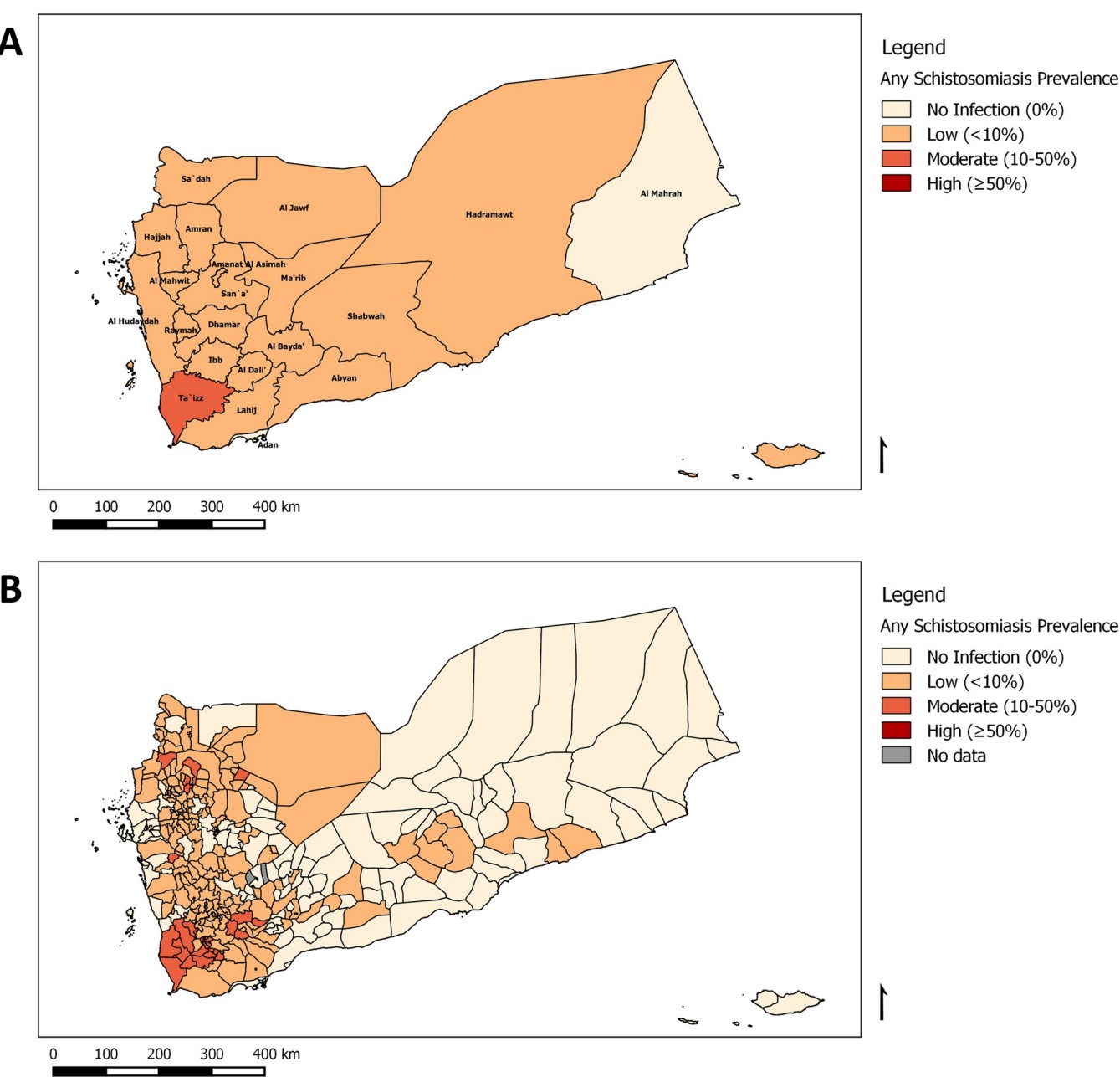

**Fig 2. Prevalence of any schistosomiasis (intestinal or urogenital) infection in Yemen, 2014.** The maps depict any schistosomiasis prevalence at (A) Governorate and (B) District level. These figures were created for this manuscript in QGIS using open source data from DIVA-GIS for the base layers. (DIVA-GIS-http://www.diva-gis.org/gdata).

However, it is important to explain that the two surveys used different methodological approaches which restricts the ability to conduct direct comparisons. The 2010 survey included demographic and ecological data from 41,381 mapped villages, including village population, number of houses, number of families, male and female population, and age. The data were combined in a geospatial model with relevant ecological covariates (rivers and other water sources, rainfall, temperature, population distribution, land cover and altitude) to map the risk of helminth transmission and infection across the country with the purpose of

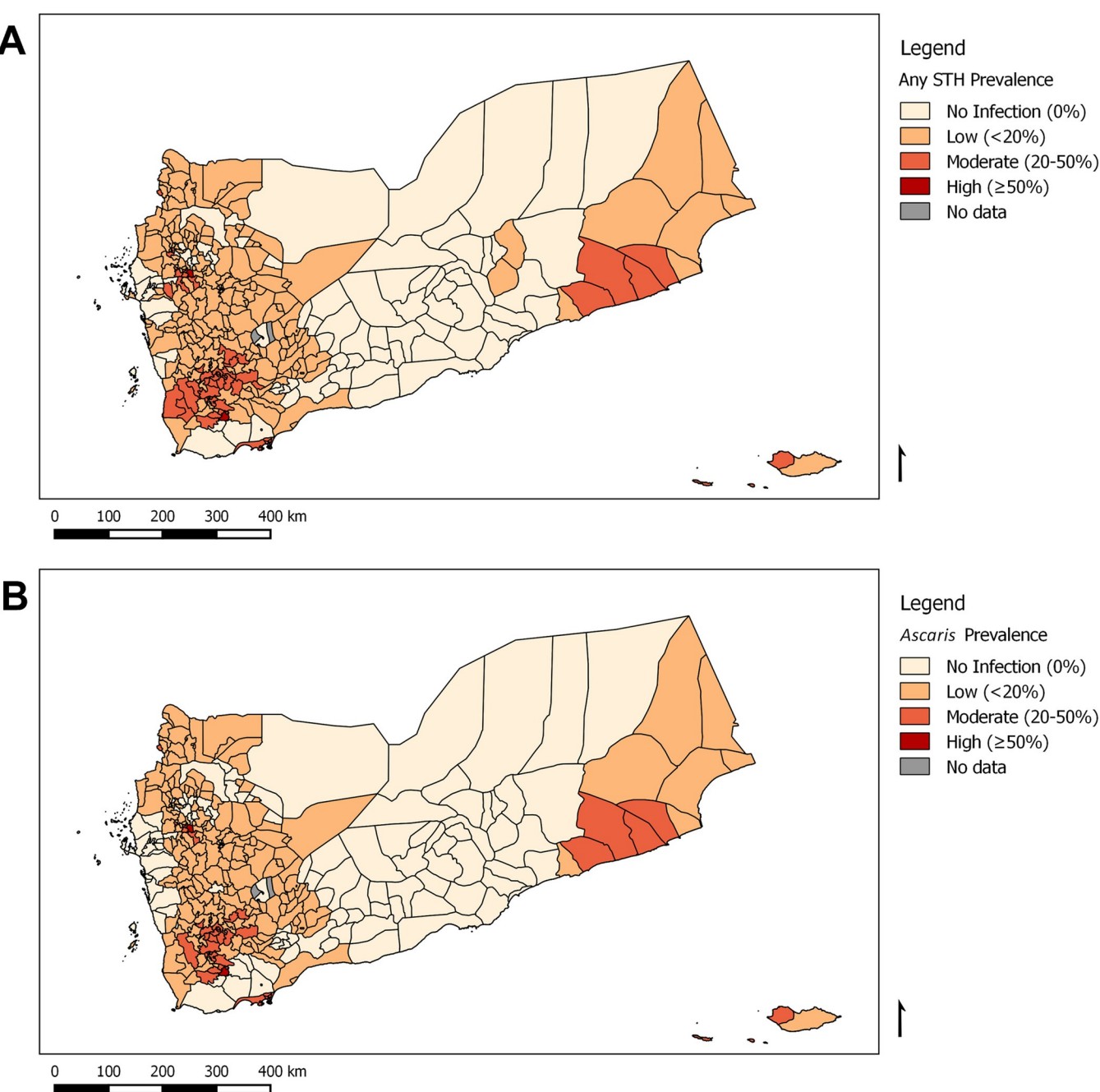

**Fig 3. District-level point prevalence of soil-transmitted helminthaisis infection in Yemen, 2014.** The maps depict (A) any soil-transmitted helminth infection and (B) *Ascaris lumbricoides* infection. These figures were created for this manuscript in QGIS using open source data from DIVA-GIS for the base layers. (DIVA-GIS-http://www.diva-gis.org/gdata).

stratifying districts according to level of risk [10]. The 2010 survey also employed a GIS-based implementation planning kit. This supported the planning process for every village, sub-district (ozla), treatment point, working team, district, and province in Yemen. The kit automated the calculation of the targeted population subgroups for treatment, the required drug amounts, and the operational and social mobilization costs at all levels. The population estimation was based on 2004 population figures for 333 districts [10].

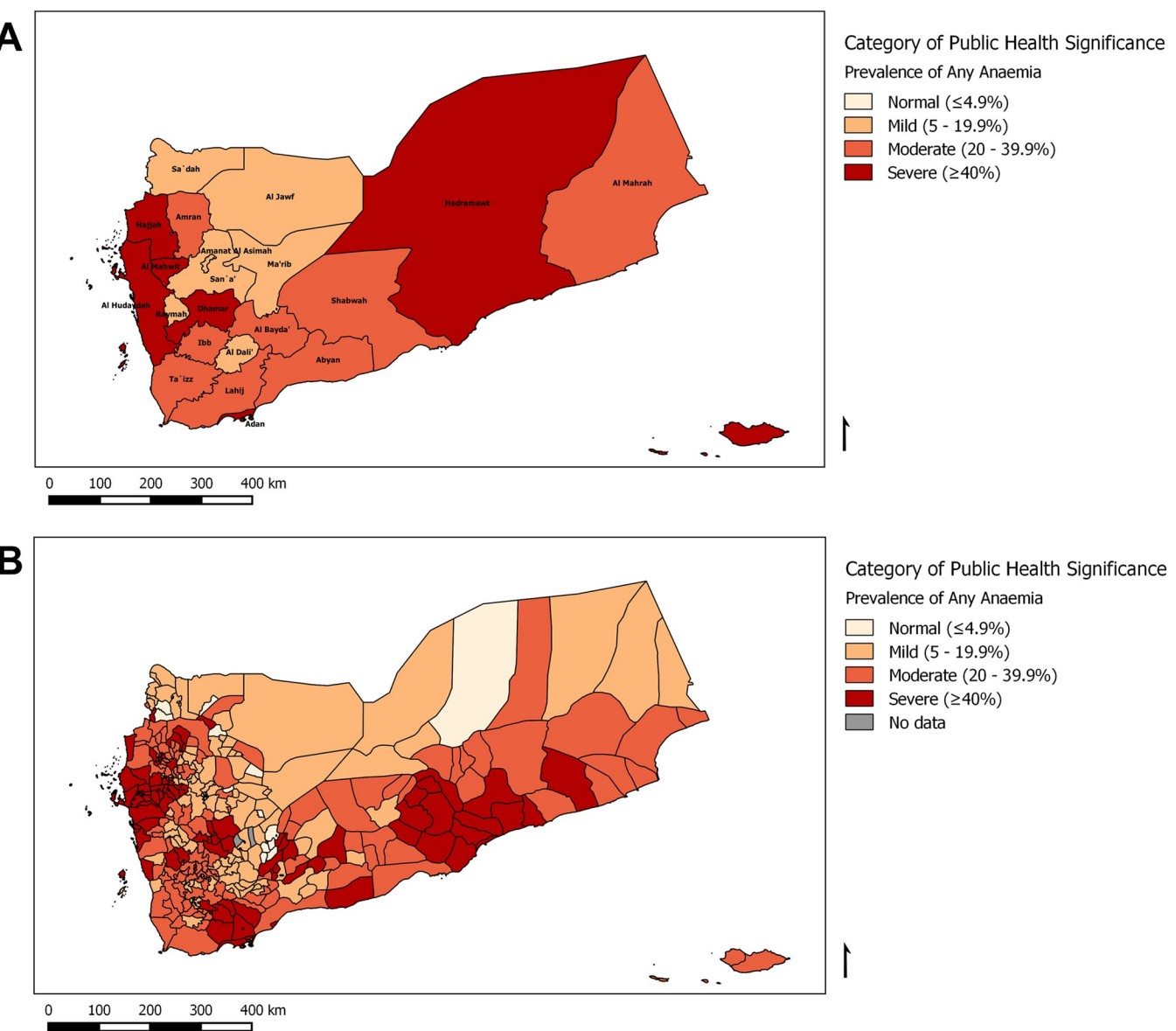

**Fig 4. Prevalence maps of the public health significance of anaemia in Yemen, 2014.** The maps display anaemia severity nationwide at (A) the governorate level and (B) the district level. These figures were created for this manuscript in QGIS using open-source data from DIVA-GIS for the base layers. (DIVA-GIS-http://www.diva-gis.org/gdata).

Despite the difference in the methods used, the findings highlight a reduction in the burden of both SCH and STH across the country after implementing the national intervention activities by the NSCP. SCH control was sporadic until 2008, when the five-year national

**Table 5. District-level distribution of anaemia in Yemen, 2014.**

| Category (prevalence [%]) | Number of districts | % of districts |
|---|---|---|
| No anaemia (≤ 4.9) | 12 | 3.6% |
| Mild (5.0–19.9) | 112 | 33.7% |
| Moderate (20.0–39.9) | 120 | 36.1% |
| Severe (≥ 40) | 88 | 26.5% |

NSCP was initiated to treat over two million school-aged children via preventive chemotherapy using anthelminthic drugs. Following the first three years of programme implementation, nine rounds of treatment were conducted with 45 million PZQ tablets and 18 million ALB tablets distributed to 13 million people. MDA was conducted using a combination of fixed and temporary sites (schools, mosques, health facilities) and mobile teams. This approach has been demonstrated to be effective in reaching the programme's key target groups of enrolled and non-enrolled school-aged children, and adults in 275 out of 333 districts based on data from 2010. In the most affected districts, individuals have received up to three doses of PZQ once per year. Much of the east of the country is effectively SCH-free (Fig 1A and 1B), with much lighter infection in many other places. However, more interventions need to be implemented in other districts, especially in the western districts (Fig 2B), toward better national control.

There was no evidence of any changes in environmental health interventions aimed at improving sanitation and safe water supply provision which could have contributed to reductions in helminth infections. Treatment alone is unlikely to lead to elimination of SCH or STH infections which is thought to also require health education, improved access to safe water, and adequate sanitation facilities in order to stop open defecation and improve personal hygiene practices [1–3, 11]; absence of these interventions could lead to infection morbidity recrudescence [37–40]. For example, lack of access to handwashing facilities in schools and in the community are likely to promote STH transmission [39, 40]. It is recommended to involve the private sector and NGOs in implementing these activities, as historically they have shown effectiveness in supporting and sustaining similar activities in other low and middle-income countries [39, 41–43].

This study also identified that anaemia is prevalent among schoolchildren in almost all Yemeni districts. Anaemia is a severe public health problem in over a quarter of districts and a moderate public health problem in a further third. Anaemia is a known sequela of various infections prevalent in the country, including SCH, STH, and malaria [14, 15] and is known to be correlated with the high rates of undernutrition (stunting and wasting) among children, especially in rural populations [13]. Few subnational studies have indicated that the prevalence of anaemia among Yemeni children under five years old range from 20.0% to 73.5% and suggest that it is more prevalent among those who have worm infestations [44, 45]. However, none of the previous national surveys had assessed the prevalence of anaemia among schoolchildren [12]. The Yemen National Health and Demographic Survey regularly assess anaemia among children under five and women of reproductive age. Lack of district anaemia data from 2010 precluded any estimation of the impact of the NSCP on the prevalence of anaemia. A Master's thesis conducted in 2018 explored the prevalence of intestinal SCH and its effect on nutritional status and anaemia among schoolchildren in two districts in Sana'a Governorate [46]. The study demonstrated that the highly focal distribution of intestinal SCH was related to prevalence of wasting and stunting in early adolescents. Anaemia is still a moderate public health problem in early female adolescents. Underweight and wasting were identified to be high among adolescent school-aged children in two governorates endemic with SCH [9, 45]. Evidence from other contexts suggest that anthelminthic treatments are effective in improving haemoglobin levels and reducing the prevalence of anaemia and undernutrition in general [42, 47].

The findings of this PM survey signal that SCH distribution is much reduced compared to the 2010 estimates. This stimulated asking questions about the possibility of shifting the programme objectives toward elimination of transmission rather than just control, in line with the WHO 2020 and 2025 goals [37, 38, 48] and the global target [47, 49, 50]. However, the end of the programme grants as well as the political instability in the country and the huge

disruption since the eruption of the conflict in 2015, which affected all sectors including education and health, are major challenges toward implementation of large-scale programmes, and hence of achieving the ambitious target of elimination of SCH in Yemen.

WHO recommends using school-based sampling to estimate the distribution of helminth infections [11, 21] and school-based deworming interventions have been demonstrated to reduce the burden of helminth infections [51]. Rapid diagnostic tests can play a role in improving the efficiency of diagnosis in field settings, particularly where access to laboratory equipment is constrained [52, 53]. However, given the continuing instability, school-based monitoring and deworming is incredibly challenging. As UNICEF reported in mid-2021, the conflict in Yemen has pushed almost two million children out of school and meant that two-thirds of Yemeni teachers have been without salaries for more than four years [54]. Children were forced to leave schools due to insecurity, school destruction, or to help families with income generation and household tasks [55]. In addition, these challenges necessitated a change in funding approach so that MDAs are an output-based payment for those Yemeni teachers. Such status affects the quality of drug distribution, consequently, MDAs reverted from a large scale into a small scale covering mainly the high and meso endemic districts on alternative years. Even though the MDA campaigns have continued to cover the enrolled and the non-enrolled school-aged children through fixed and community-based treatment, pre-campaign impact evaluation studies have completely ceased since 2015. Consequently, there have not been any more regular parasitological assessment surveys including anaemia and anthropometric measurements on school-aged children [56]. These are clear barriers to controlling SCH and STH and improving anaemia and other health challenges in Yemen.

## 5. Conclusions and recommendations

This study provided a comprehensive mapping of SCH, STH, and anaemia in Yemen. Anaemia was identified as a severe or moderate public health problem in at least a quarter and a third, respectively, of Yemeni districts. This reinforces the need for interventions to reduce the burden of anaemia across the country. Although malnutrition and helminth infections are the most suspected causes of anaemia, confirmatory research is needed in this regard.

The results show reduced levels of both SCH and STH infections across all Yemen districts compared to a previous survey in 2010 after implementing national intervention activities. However, this status cannot be guaranteed after ending of the programme grant and the political instability in the country due to the war which has been ongoing since 2014. There is a need to restore the programme's full functioning capacity to protect gains in infection levels and prevent recrudescence of infection. Without comprehensive community awareness, behaviour change interventions, and significant improvement of the provision of safe water and adequate sanitary facilities, it will be challenging to interrupt the SCH and STH transmission cycle and eliminate these infections across the country. Combating the burden of STH and SCH and their consequences requires a holistic intervention approach that should involve different stakeholders including community members and local authorities.

Although the distribution of infection will undoubtedly have changed somewhat since this survey was conducted in 2014, these data provide a strong platform and most recent disease distribution data for planning a sustained national control and elimination programme when the security situation allows. Future studies should also explore the impact of the programme on anaemia as well as nutritional status and educational achievement of children. This would enhance appropriate resource allocation for controlling neglected tropical diseases and their sequelae in Yemen.

## Supporting information

**S1 Fig. Point prevalence map of *Trichuris* infection in Yemen, 2014.** This figure was created for this manuscript in QGIS using open-source data from DIVA-GIS for the base layers. (DIVA-GIS-http://www.diva-gis.org/gdata).
(TIF)

**S2 Fig. Point prevalence map of Hookworm infection in Yemen, 2014.** This figure was created for this manuscript in QGIS using open-source data from DIVA-GIS for the base layers. (DIVA-GIS-http://www.diva-gis.org/gdata).
(TIF)

## Acknowledgments

First and foremost, the authors would like to thank the children of Yemen who took part in the study. The authors acknowledge everyone who supported or participated in the implementation process of this survey, field data, and sample collection, including team members, laboratory technicians, enumerators as well as education, school authorities.

## Author Contributions

**Conceptualization:** Dhekra Amin Annuzaili, Hani Farouk El-Talabawy, Abdulsalam M. Al-Mekhlafi, Samira Al-Eryani, Abdulhakim Ali Alkohlani, Albis Francesco Gabrielli, Riadh Ben-Ismail, Sami Alhaidari, Adel Muaydh, Rasheed Alshami, Magid Al Gunaid, Alaa Hamed, Nehad Kamel, Fiona Fleming, Michael Duncan French.

**Data curation:** Nur Alia Johari, Hani Farouk El-Talabawy, Abdulsalam M. Al-Mekhlafi, Samira Al-Eryani, Fiona Fleming, Michael Duncan French.

**Formal analysis:** Nur Alia Johari, Dhekra Amin Annuzaili, Hani Farouk El-Talabawy, Albis Francesco Gabrielli, Fiona Fleming, Michael Duncan French.

**Funding acquisition:** Dhekra Amin Annuzaili, Abdulhakim Ali Alkohlani, Magid Al Gunaid, Alaa Hamed, Nehad Kamel, Karen Palacio, Michael Duncan French.

**Investigation:** Dhekra Amin Annuzaili, Abdulsalam M. Al-Mekhlafi, Samira Al-Eryani, Alaa Hamed, Nehad Kamel, Fiona Fleming, Michael Duncan French.

**Methodology:** Dhekra Amin Annuzaili, Hani Farouk El-Talabawy, Abdulsalam M. Al-Mekhlafi, Samira Al-Eryani, Albis Francesco Gabrielli, Riadh Ben-Ismail, Sami Alhaidari, Adel Muaydh, Fiona Fleming, Michael Duncan French.

**Project administration:** Dhekra Amin Annuzaili, Samira Al-Eryani, Abdulhakim Ali Alkohlani, Albis Francesco Gabrielli, Sami Alhaidari, Adel Muaydh, Rasheed Alshami, Magid Al Gunaid, Nehad Kamel, Karen Palacio, Michael Duncan French.

**Resources:** Dhekra Amin Annuzaili, Hani Farouk El-Talabawy, Maryam Ba-Break, Riadh Ben-Ismail, Alaa Hamed, Nehad Kamel, Karen Palacio, Michael Duncan French.

**Software:** Nur Alia Johari, Hani Farouk El-Talabawy, Abdulsalam M. Al-Mekhlafi, Samira Al-Eryani, Michael Duncan French.

**Supervision:** Dhekra Amin Annuzaili, Abdulsalam M. Al-Mekhlafi, Samira Al-Eryani, Sami Alhaidari, Adel Muaydh, Rasheed Alshami, Michael Duncan French.

**Validation:** Dhekra Amin Annuzaili, Abdulsalam M. Al-Mekhlafi, Samira Al-Eryani, Albis Francesco Gabrielli, Riadh Ben-Ismail, Karen Palacio, Fiona Fleming, Michael Duncan French.

**Visualization:** Nur Alia Johari, Dhekra Amin Annuzaili, Maryam Ba-Break, Abdulsalam M. Al-Mekhlafi, Albis Francesco Gabrielli, Fiona Fleming, Michael Duncan French.

**Writing – original draft:** Nur Alia Johari, Dhekra Amin Annuzaili, Abdulsalam M. Al-Mekhlafi, Michael Duncan French.

**Writing – review & editing:** Nur Alia Johari, Dhekra Amin Annuzaili, Hani Farouk El-Talabawy, Maryam Ba-Break, Samira Al-Eryani, Abdulhakim Ali Alkohlani, Albis Francesco Gabrielli, Michael Duncan French.

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
