## [Decision Letter · Decision Letter 0]

28 Jun 2021

Dear Dr Annuzaili,

Thank you very much for submitting your manuscript " National Mapping of Schistosomiasis, Soil-Transmitted Helminthiasis and Anemia  in Yemen: Towards Better National Control and Elimination " for consideration at PLOS Neglected Tropical Diseases. As with all papers reviewed by the journal, your manuscript was reviewed by members of the editorial board and by several independent reviewers. In light of the reviews (below this email), we would like to invite the resubmission of a significantly-revised version that takes into account the reviewers' comments. 

We cannot make any decision about publication until we have seen the revised manuscript and your response to the reviewers' comments. Your revised manuscript is also likely to be sent to reviewers for further evaluation.

Sincerely,

Alberto Novaes Ramos Jr

Associate Editor

Marco Coral-Almeida

Deputy Editor

Reviewer's Responses to Questions

**Key Review Criteria Required for Acceptance?**

**Methods**

-Are the objectives of the study clearly articulated with a clear testable hypothesis stated?

-Is the study design appropriate to address the stated objectives?

-Is the population clearly described and appropriate for the hypothesis being tested?

-Is the sample size sufficient to ensure adequate power to address the hypothesis being tested?

-Were correct statistical analysis used to support conclusions?

-Are there concerns about ethical or regulatory requirements being met?

Reviewer #1: see summary and general comments

Reviewer #2: Find my comments attached

Reviewer #3: See general comments.

Reviewer #4: See attached

**Results**

-Does the analysis presented match the analysis plan?

-Are the results clearly and completely presented?

-Are the figures (Tables, Images) of sufficient quality for clarity?

Reviewer #1: see summary and general comments

Reviewer #2: Find my comments attached

Reviewer #3: See general comments.

Reviewer #4: See attached

**Conclusions**

-Are the conclusions supported by the data presented?

-Are the limitations of analysis clearly described?

-Do the authors discuss how these data can be helpful to advance our understanding of the topic under study?

-Is public health relevance addressed?

Reviewer #1: see summary and general comments

Reviewer #2: Find my comments attached

Reviewer #3: See general comments.

Reviewer #4: See attached

**Editorial and Data Presentation Modifications?**

Reviewer #1: (No Response)

Reviewer #2: (No Response)

Reviewer #3: Minor Comments:

1) Lines 109-11: When were the 45 million tablets distributed to 13 million people? After the 2014 survey? Between 2010 and 2014?

2) Line 256: for clarity, please define “endemic”.

3) Lines 260-262: were there districts that warranted STH treatment that did not require PZQ treatment? Were they treated?

4) Fig. 3: as above, were data comparable for 2010 and 2014?

5) Lines 290-291: “About a fifth (22%) of Yemeni districts have Ascaris infections among 20%-40% of school children.” Fig. 4 uses the WHO thresholds of 20 and 50%. These numbers should agree.

Reviewer #4: See attached

**Summary and General Comments**

Reviewer #1: The article is important since is documenting the epidemiological situation od Schistosomiasis and STH infection in Yemen in 2014, is well written and in my opinion merits publication.

I have only minor changes to request:

1- one point is not completely clear: the author mention that the one conducted in 2014 wass the first national survey conducted in Yemen but then in figure compare the endemicity in 2010 and 2014; so 2010 data are present? If not how the classification of each district in the different classes of endemicity was done?

2- In line 386-388 the authors mention the programme was successful in control the SCH related morbidity but do not provided an indicator for that claim.

The morbidity of SCH (and STH infections) is measured by WHO as the prevalence of infections of moderate heavy intensity (as infection of light intensity are causing minimal morbidity) it would therefore be important (if the data are available) to report the prevalence of infection of moderate/ heavy intensity for SCH (and STH)(and present it in a map.

This will provide a clear idea of the morbidity due to these infection present in Yemen in 2014.

Reviewer #2: Find my comments attached

Reviewer #3: In this manuscript, Johari and colleagues report the results of a large scale mapping exercise for schistosomiasis and soil transmitted helminths that was conducted in Yemen in 2014. The impressive scale of the survey– more than 80,000 children tested – makes these data especially noteworthy. It is a pity that there was such a delay in the reporting of this information and that the war has undermined the country’s ability to provide MDA.

I have a few concerns about this paper. First, as noted in the discussion, the authors are trying to make the case that the 2014 survey results document “substantial and significant reduction in the burden of both SCH and STH.” However, except for the information presented in Fig. 3, the reader has no access to information about the 2010 survey: how extensive was it, were the same districts tested, were the same age groups? I did not see these results referenced. If the comparison of the 2010 and 2014 results is a central conclusion of the paper, more information should be provided about the 2010 survey. 

I don’t understand why this paper is using thresholds for SCH that are different from those used by WHO. I understand the desire to be more conservative, but the programmatic decision can be made separately from the presentation of the data here. 

Given the scale of the tragedy that has unfolded in Yemen, I think it is important for the paper to refer to this more specifically. Are children in school? Is MDA ongoing anywhere? How has the disruption in services affected children in terms of their growth and anemia status? Are the “gains” being reversed?

Reviewer #4: See attached

PLOS authors have the option to publish the peer review history of their article (what does this mean?). If published, this will include your full peer review and any attached files.

Reviewer #1: No

Reviewer #2: Yes: Jean T. Coulibaly

Reviewer #3: No

Reviewer #4: No
---

## [Decision Letter · Decision Letter 1]

4 Oct 2021

Dear Dr Annuzaili,

Thank you very much for submitting your manuscript "National Mapping of Schistosomiasis, Soil-Transmitted Helminthiasis and Anemia  in Yemen: Towards Better National Control and Elimination" for consideration at PLOS Neglected Tropical Diseases. As with all papers reviewed by the journal, your manuscript was reviewed by members of the editorial board and by several independent reviewers. In light of the reviews (below this email), we would like to invite the resubmission of a significantly-revised version that takes into account the reviewers' comments. 

We cannot make any decision about publication until we have seen the revised manuscript and your response to the reviewers' comments. Your revised manuscript is also likely to be sent to reviewers for further evaluation.

Sincerely,

Alberto Novaes Ramos Jr

Associate Editor

Marco Coral-Almeida

Deputy Editor

Reviewer's Responses to Questions

**Key Review Criteria Required for Acceptance?**

**Methods**

-Are the objectives of the study clearly articulated with a clear testable hypothesis stated?

-Is the study design appropriate to address the stated objectives?

-Is the population clearly described and appropriate for the hypothesis being tested?

-Is the sample size sufficient to ensure adequate power to address the hypothesis being tested?

-Were correct statistical analysis used to support conclusions?

-Are there concerns about ethical or regulatory requirements being met?

Reviewer #3: (No Response)

Reviewer #4: The authors have satisfactorily addressed the my original comments in regard to methods with the exception of the use of baseline (please see my comments in the summary/general comments section below).

**Results**

-Does the analysis presented match the analysis plan?

-Are the results clearly and completely presented?

-Are the figures (Tables, Images) of sufficient quality for clarity?

Reviewer #3: See below

Reviewer #4: The authors have satisfactorily addressed the my original comments in regard to results.

**Conclusions**

-Are the conclusions supported by the data presented?

-Are the limitations of analysis clearly described?

-Do the authors discuss how these data can be helpful to advance our understanding of the topic under study?

-Is public health relevance addressed?

Reviewer #3: See below

Reviewer #4: (No Response)

**Editorial and Data Presentation Modifications?**

Reviewer #3: Minor Comments:

1) Lines 121-22: The authors have now provided the years when the 45 million tablets distributed to 13 million people; however, it’s hard to understand how these numbers relate to the number of years of treatment and coverage. 

2) Line 204-205: “The results were then cross-checked before being to the SCI.” A word is missing here.

Reviewer #4: I suggest moving the newly added figures in the discussion section to the supplementary folder. The discussion is not the place for presenting this data visually.

**Summary and General Comments**

Reviewer #3: In this manuscript, Johari and colleagues report the results of a large scale mapping exercise for schistosomiasis and soil transmitted helminths that was conducted in 2014. The scale of this survey was impressive – more than 80,000 children tested – makes these data especially noteworthy. It is a pity that there was such a delay in the reporting of this information and that the war has undermined the ability to provide MDA.

I still have a few concerns about this paper. First, as noted in the discussion, the authors are trying to make the case that the 2014 survey results document “substantial and significant reduction in the burden of both SCH and STH.” Despite some additions to the text, the reader still doesn’t have access to the 2010 survey results or even an adequate description of the methodology. If the comparison of the 2010 and 2014 results is a central conclusion of the paper, more information should be provided about the 2010 survey. 

I don’t understand why this paper is using thresholds for SCH that are different from those used by WHO. I understand the desire to be more conservative, but the programmatic decision can be made separately from what is reported here. 

Given the scale of the tragedy that has unfolded in Yemen, I think it is important for the paper to refer to this more specifically. Are children in school? Is MDA ongoing anywhere? How has the disruption in services impacted children in terms of their growth and anemia status? Are the “gains” being reversed?

Reviewer #4: The authors have done well to address all the reviewer comments so thoroughly. Although I agree with the authors response to my comment (#3) regarding the use of previous mapping data to highlight the progress made, I still disagree with the use of the term 'baseline' for this earlier data. I strongly suggest avoiding the use of baseline and follow up as it suggests a very different type of analysis than what has been presented in this manuscript. Rather, the data could be referred to by the year, e.g., 2010 survey and 2014 survey.

PLOS authors have the option to publish the peer review history of their article (what does this mean?). If published, this will include your full peer review and any attached files.

Reviewer #3: No

Reviewer #4: No
---

## [Decision Letter · Decision Letter 2]

14 Dec 2021

Dear Dr Annuzaili,

We are pleased to inform you that your manuscript 'National Mapping of Schistosomiasis, Soil-Transmitted Helminthiasis and Anemia  in Yemen: Towards Better National Control and Elimination' has been provisionally accepted for publication in PLOS Neglected Tropical Diseases.

Best regards,

Alberto Novaes Ramos Jr

Associate Editor

Marco Coral-Almeida

Deputy Editor

Reviewer's Responses to Questions

**Key Review Criteria Required for Acceptance?**

**Methods**

-Are the objectives of the study clearly articulated with a clear testable hypothesis stated?

-Is the study design appropriate to address the stated objectives?

-Is the population clearly described and appropriate for the hypothesis being tested?

-Is the sample size sufficient to ensure adequate power to address the hypothesis being tested?

-Were correct statistical analysis used to support conclusions?

-Are there concerns about ethical or regulatory requirements being met?

Reviewer #1: objective of the study is clear and method well explained

Reviewer #3: (No Response)

**Results**

-Does the analysis presented match the analysis plan?

-Are the results clearly and completely presented?

-Are the figures (Tables, Images) of sufficient quality for clarity?

Reviewer #1: Results are well reported

Reviewer #3: (No Response)

**Conclusions**

-Are the conclusions supported by the data presented?

-Are the limitations of analysis clearly described?

-Do the authors discuss how these data can be helpful to advance our understanding of the topic under study?

-Is public health relevance addressed?

Reviewer #1: The conclusions are supported by the data resented

Reviewer #3: (No Response)

**Editorial and Data Presentation Modifications?**

Reviewer #1: (No Response)

Reviewer #3: (No Response)

**Summary and General Comments**

Reviewer #1: I think the paper report important epidemiological data on schistosomiasis and STH and should be published in the present form.

Reviewer #3: (No Response)

PLOS authors have the option to publish the peer review history of their article (what does this mean?). If published, this will include your full peer review and any attached files.

Reviewer #1: No

Reviewer #3: No

---

## [Editor Report · Decision Letter 3]

17 Feb 2022

Dear Dr Annuzaili,

We are pleased to inform you that your manuscript 'National Mapping of Schistosomiasis, Soil-Transmitted Helminthiasis and Anaemia  in Yemen: Towards Better National Control and Elimination' has been provisionally accepted for publication in PLOS Neglected Tropical Diseases.

Best regards,

Alberto Novaes Ramos Jr

Associate Editor

Marco Coral-Almeida

Deputy Editor

The authors have included in the revised version of the manuscript an updated ethics statement. These adjustments satisfactorily address the ethical issues previously indicated.

---

## [Editor Report · Acceptance letter]

4 Mar 2022

Dear Dr Annuzaili,

We are delighted to inform you that your manuscript, " National Mapping of Schistosomiasis, Soil-Transmitted Helminthiasis and Anaemia  in Yemen: Towards Better National Control and Elimination ," has been formally accepted for publication in PLOS Neglected Tropical Diseases.

Best regards,

Shaden Kamhawi

co-Editor-in-Chief

Paul Brindley

co-Editor-in-Chief
